# Organismal Fructose Metabolism in Health and Non-Alcoholic Fatty Liver Disease

**DOI:** 10.3390/biology9110405

**Published:** 2020-11-18

**Authors:** Shea Skenderian, Grace Park, Cholsoon Jang

**Affiliations:** 1Department of Molecular & Cell Biology, University of California Berkeley, Berkeley, CA 94720, USA; sheaskenderian@berkeley.edu; 2Department of Biological Chemistry, University of California Irvine, Irvine, CA 92697, USA; shinhyepark9@berkeley.edu

**Keywords:** fructose, sucrose, liver, small intestine, gut microbiota, lipogenesis, inflammation, fatty liver disease, NAFLD, ketohexokinase

## Abstract

**Simple Summary:**

The consumption of dietary fructose as sugar and high fructose corn syrup (HFCS), which is abundant in soft drinks, has markedly increased. This trend has been accompanied by an alarmingly increased incidence of non-alcoholic fatty liver disease (NAFLD). Recent studies using disease animal models such as mice and rats have revealed several important aspects of how our body handles fructose, especially when it is consumed in a large amount. Moreover, not only our bodily organs, but also microorganisms residing in the gut, have been shown to actively digest fructose and contribute to NAFLD. In this article, we summarize recent progress in our understanding of fructose metabolism at the organismal level. This review assembles scientific evidence that encourages the public to avoid an excess intake of fructose to prevent NAFLD and suggests potential drug targets to treat the disease.

**Abstract:**

NAFLD has alarmingly increased, yet FDA-approved drugs are still lacking. An excessive intake of fructose, especially in liquid form, is a dietary risk factor of NAFLD. While fructose metabolism has been studied for decades, it is still controversial how fructose intake can cause NAFLD. It has long been believed that fructose metabolism solely happens in the liver and accordingly, numerous studies have investigated liver fructose metabolism using primary hepatocytes or liver cell lines in culture. While cultured cells are useful for studying detailed signaling pathways and metabolism in a cell-autonomous manner, it is equally important to understand fructose metabolism at the whole-body level in live organisms. In this regard, recent in vivo studies using genetically modified mice and stable isotope tracing have tremendously expanded our understanding of the complex interaction between fructose-catabolizing organs and gut microbiota. Here, we discuss how the aberrant distribution of fructose metabolism between organs and gut microbiota can contribute to NAFLD. We also address potential therapeutic interventions of fructose-elicited NAFLD.

## 1. Introduction

Technical breakthroughs in the food industry have impacted our diets in many ways over the last century. This has coincided with a dramatic increase in the incidence of metabolic and cardiovascular disease. One of the most marked changes is the highly increased consumption of dietary fructose as sugar and high fructose corn syrup (HFCS), especially in liquid form such as soft drinks. Fructose is abundant in natural products, including fruits, honeys, and vegetables, which are regarded as healthy. Historically, fructose was purified from sugar cane or sugar beets and served as a sweetener for a variety of drinks (e.g., British black tea), but its high price only allowed consumption by the rich. However, due to the technological development that enabled the mass production of fructose from corn with an enzymatic reaction (glucose isomerase that converts glucose to fructose), the price of HFCS and sugar dropped dramatically. Additionally, past research that focused on revealing the detrimental effects of fat and cholesterol on cardiovascular disease encouraged the public to reduce their fat and cholesterol consumption, but ironically, this trend increased the amount of fructose in processed foods as a replacement for fat (e.g., low-fat yogurts contain a large amount of fructose). Animal feeding studies and human epidemiological evidence have indicated that there is a causal relationship between fructose consumption and obesity, type 2 diabetes, non-alcoholic fatty liver disease (NAFLD), and cardiovascular disease. Many great reviews written by others have described fructose metabolism at cellular levels and the mechanistic regulation of fructose-related enzymes. Therefore, in this review, we will focus on fructose metabolism at the organismal level, including the role of various fructose-catabolizing organs and gut microbiota in health and diseases, especially NAFLD. Finally, we will discuss therapeutic strategies for treating fructose-induced NAFLD.

## 2. Intestinal Fructose Absorption and Metabolism

Fructose can be endogenously synthesized in the body via the polyol pathway, while the majority of fructose comes from the diet [1]. The dietary fructose intake has markedly increased over the past century, and has now reached almost 20% of the total carbohydrate intake. Despite having exactly the same molecular formula (C_6_H_12_O_6_) as glucose, fructose requires very distinct transporters for intestinal absorption: Glut5 and Glut2 (Figure 1) [2,3,4]. The importance of these transporters has been demonstrated by genetically modified mice and rare hereditary mutations found in humans. Glut2, which is expressed in the intestine, liver, and kidneys, was firstly identified as a fructose transporter [5]. Later, Glut2 was shown to be located on the basolateral side of the intestinal epithelial cells and thereby mediates the transport of intra-cellular fructose to the blood stream [4]. Importantly, Glut2 whole-body knockout mice show only mildly decreased fructose absorption [4], indicating the potential existence of another fructose transporter. 

Glut5 is most highly expressed in the small intestine and kidneys, but hardly expressed in other organs, including the liver. Glut5 is located on the apical side of the intestinal epithelial cells and mediates the active transport of fructose from the intestinal lumen into the epithelial cells [2]. Glut5 whole-body knockout mice survive without any defects under typical chow diets, but they show lethal phenotypes upon fructose feeding [2]. This severe fructose intolerance is accompanied by a distended large intestine due to unabsorbed fructose and fluid retention. These phenotypes are reminiscent of the symptoms in hereditary fructose-intolerant patients, although these patients’ symptoms are related to aldolase B deficiency rather than GLUT5 mutation [6]. 

It is evident that there is interplay between intestinal fructose absorption and metabolic dysregulation. Children with NAFLD have higher levels of fructose absorption and lower levels of serum fructose than their lean counterparts [7]. In mice, GLUT5 deficiency causes hepatic steatosis, in addition to intestinal distention and fluid retention [8]. Unabsorbed fructose in the intestine can increase the intestinal permeability by the induction of epithelial stress and barrier degradation. Alleviation of these symptoms by antibiotics treatment suggests the role of fructose-induced microbial dysbiosis in hepatic steatosis [9]. In support of this model, increased levels of pro-inflammatory cytokines such as TNFα and endotoxins have been found in the portal blood [9,10]. This microbial endotoxin reaching the liver is implicated in hepatic triglyceride accumulation and inflammation, which are the major symptoms of NAFLD [10].

Fructose inside intestinal epithelial cells can have another important fate: Instead of being transported into portal circulation, it can be catabolized by ketohexokinase (KHK) within the intestinal epithelial cells, which phosphorylates fructose into fructose-1-phosphate (F1P) [11,12,13,14,15]. KHK has two splice variants—KHK-C and KHK-A—of which the former is greatly expressed in the liver, small intestine, pancreas, and kidney [12,16]. KHK-A is more broadly expressed in multiple tissues [12,16]. KHK-C has an affinity for fructose that is ten times greater than that of KHK-A, making it of primary importance in fructolysis [12]. So far, KHK-C-specific whole-body knockout mice have not been reported, but KHK-C/A whole-body double knockout mice exhibited completely blunted fructose metabolism, with the excretion of most fructose by urine [16]. On the other hand, KHK-A whole-body knockout mice did not show this phenotype due to the intact KHK-C. Therefore, based on its high Km and no overt knockout phenotype, it is suggested that KHK-A may have substrates other than fructose [17]. However, given that KHK-C-specific knockout mice only showed a 50% reduction of intestinal fructose catabolism in the intestine [13], KHK-A seems to play important roles in catabolizing fructose, at least in the intestine, where fructose concentrations are higher than KHK-A’s Km. In fact, intestine-specific KHK-C/A double-knockout mice exhibited completely blunted intestinal fructose catabolism [14].

How is the intestinal fructose metabolism regulated? Mavrias et al. demonstrated increased intestinal KHK expression in rats within 3 days of initial fructose exposure [15]. Patel et al. used KHK-C/A knockout mice or Glut5 knockout mice to prove that both intestinal fructose absorption and catabolism are essential for the intestinal induction of fructose catabolic genes (aldolase B and trios kinase) and gluconeogenic genes (fructose-1,6-bisphosphatase and glucose-6-phosphatase) [2]. Additionally, Kim et al. proved that this transcriptional regulation by fructose is mediated by the transcription factor, carbohydrate-response element-binding protein (ChREBP), by generating intestine-specific ChREBP knockout mice [18]. These mice displayed completely suppressed Glut5, KHK, and other fructolytic and gluconeogenic enzyme expression, even after chronic fructose feeding. 

Compared to KHK, downstream enzymes specific for fructose catabolism are less understood at the organismal level. Aldolase B is responsible for cleaving F1P to generate glyceraldehyde and dihydroxyacetone phosphate (Figure 1) [19]. Then, trios kinase converts glyceraldehyde to glyceraldyde-3-phosphate [20], which enters glycolysis. Like Glut5 knockout mice, aldolase B whole-body knockout mice experience a lethal phenotype upon fructose feeding with severe hepatic fat accumulation and fibrosis, which phenocopies hereditary fructose intolerance in humans with aldolase B mutations [21]. After being fed fructose, these mice showed highly accumulated F1P in the liver, confirming that aldolase B is the sole enzyme required for F1P degradation. Importantly, unlike Glut 5 or ChREBP knockout mice, which likely die due to intestinal extension, the lethality and pathological phenotypes of aldolase B knockout mice were largely rescued by blocking KHK activity [22]. This indicates that F1P buildup is the major cause of detrimental phenotypes. This study thus provided important insights into the potential of treating aldolase B-mutated patients with KHK inhibitors, which are currently under clinical trials. However, generalization from animal to human models may serve to be an ineffective predictor of toxicity and dosage. Potential off-target effects and pharmacokinetics altered by genetic variance should be considered [23]. More studies with intestine-specific knockout mice of aldolase B, trios kinase, or other fructose-related enzymes are required to fully understand the patho-physiological roles of intestinal fructose metabolism.

Several critical questions remain unanswered. It is still controversial which fructose metabolite(s) is responsible for activating ChREBP in the intestine. Additionally, there is a need for further research regarding which metabolic step(s) regulates the quantity of fructose for transport versus that for catabolism. One hint that KHK controls this process was obtained from intestine-specific KHK-C knockout or transgenic mice, which showed increased or reduced fructose transport to the portal circulation [13,14]. However, because intestinal KHK-C deletion also reduces fructose absorption, data interpretation must be conducted with caution. Finally, it is crucial to determine to what extent the findings in animal models reflect fructose metabolism in humans.

## 3. Hepatic Fructose Metabolism and NAFLD

Fructose feeding experiments in rodents have consistently indicated that a chronic intake of high-dose fructose causes hyperlipidemia, insulin resistance, and NAFLD. However, a meta-analysis of human epidemiological studies gave ambiguous results due to variant genetic and environmental factors [24,25,26,27,28,29,30,31]. One point of contention is how the source of fructose differentially affects metabolic consequences. Several groups have studied the effect of naturally occurring fructose in fruits and vegetables on NAFLD. Tajima et al. analyzed a dietary questionnaire of 977 men and 1467 women ranging from 40 to 69 years of age, and found that fruit but not vegetable intake was inversely correlated with NAFLD [32]. Upon adjusting the results for body mass index (BMI), however, this correlation disappeared. It is possible that fruit or vegetables may potentially block high fructose consequences due to its chemical makeup, which includes minerals, antioxidants, polyphenols, and vitamins [33]. However, this assumption has not yet been experimentally studied.

Another controversy has arisen over how excessive fructose consumption induces NAFLD. One of the theories suggests that fructose induces obesity and insulin resistance by providing high numbers of calories, thereby driving NAFLD [31,34]. In this regard, fructose is not special, but similar to glucose as a high energy carbohydrate. Johnston et al. provided healthy, overweight men with a high-fructose or high-glucose diet before an examination of the serum and liver triglyceride [34]. Each of the subjects was assigned a diet either high in fructose or high in glucose, and then consumed a scheduled diet: An isocaloric stretch of 2 weeks; a washout stretch of 6 weeks; and then a hypercaloric stretch of 2 weeks. Both groups showed no statistically significant increase in serum or liver triglyceride levels during the isocaloric stretch. However, both groups displayed increased body weights and liver triglyceride levels during the hypercaloric stretch. The study concluded that there is no difference between fructose and glucose for the induction of NAFLD. The data suggest that it was simply the overfeeding of macronutrients that led to weight gain and increased hepatic fat accumulation. 

However, a competing theory argues that fructose is unique because it induces hepatic lipogenic genes much more potently than glucose. Indeed, high-fructose feeding in mice (30% (*w*/*v*) fructose in the drinking water or 30% fructose or 60% sucrose (% kcal) in chow) resulted in more severe NAFLD phenotypes compared to isocaloric high-glucose feeding [35,36,37]. The explanation for this difference is based on the notion that glucose and fructose metabolism are quite different [38,39,40]. Once glucose is absorbed by the small intestine, it mostly bypasses the liver and is distributed to the whole body, including organs that avidly take up glucose (e.g., skeletal muscle, adipose tissues, and the brain). On the other hand, fructose that reaches the liver is almost completely absorbed by the liver and thus fructose barely reaches other organs. This difference in organ distribution is associated with the distinct regulation mechanisms between glycolysis and fructolysis. Unlike glycolysis, which is allosterically regulated at the step of phosphofructokinases (PFKs), fructolysis bypasses this critical regulation step [11,12]. Therefore, the rapid fructose catabolism in hepatocytes induces the ChREBP-dependent transcriptional activation of lipogenic enzymes. Additionally, efficient intestinal glucose absorption prevents glucose from reaching the large intestine. However, fructose readily does so, and the colonic gut microbiota convert this fructose to various metabolites, including acetate that feeds hepatic lipogenesis without overt regulation [41]. The role of microbiota in fructose-elicited hepatic lipogenesis and NAFLD will be further discussed in a later section. In sum, the fundamental difference in the bodily handling of glucose and fructose indicates that fructose is particularly hepatotoxic. 

Additionally, fructose exerts other specific effects on the liver compared to glucose. Due to the rapid fructose catabolism in the liver by KHK-mediated phosphorylation, this can deplete ATP and subsequently induces the AMP deaminase (AMPD)-dependent purine degradation pathway [42,43,44]. This leads to an increase in uric acid production [42]. Consequently, high amounts of uric acid in the hepatocytes activate mitochondrial oxidative stress, which in turn results in fatty acid synthesis [44]. Because this pathway was studied in cultured hepatocytes with supra-physiological fructose concentrations, it will be important to confirm this finding in vivo. Nevertheless, high fructose consumption is linked to hyperuricemia in humans, which is also closely associated with NALFD [45]. However, the organ source of this circulating uric acid is still unknown. Moreover, the causal relationship between increased uric acid and NAFLD is unclear. Studying this pathway in rodents is complicated because rodents have a functional uricase enzyme that degrades uric acid, whereas humans lost this enzyme during evolution [46]. Given that rodents develop NAFLD upon high-fructose feeding, despite the fact that they efficiently clear uric acid, the causal relationship between hyperuricemia and NAFLD requires further investigation. Moreover, knockout mice of urate oxidase (encoding uricase) do not develop NAFLD, although they develop hyperuricemia [47]. Therefore, further studies are required to better understand the link between hyperuricemia and NAFLD.

It is also likely that chronic fructose exposure activates various other hepatic pathways to trigger NAFLD, such as the hepatic accumulation of free cholesterol [48]. However, the effect of fructose on cholesterol synthesis is still controversial [49,50]. Extended fructose feeding has also been linked to the development of insulin resistance from mitochondrial and ER stress. Chronic fructose feeding has also been associated with inflammation, whether by the dysbiosis of microbiota or direct action by inflammation-inducing metabolites [51]. 

While the chronic effect of fructose feeding on NAFLD has been well-studied in rodents, most human studies rely on short-term feeding. As such, the results of these studies may not be indicative of the full metabolic picture, as the liver may not yet have adjusted to dietary changes, especially when these results are applied to progressive disease like NAFLD. Epidemiological studies may thus give valuable insights on the impact of chronic fructose consumption. In one such study, patients with steatohepatitis that self-reported higher fructose consumption also had severer liver fibrosis [52].

## 4. Relationship between Intestinal and Hepatic Fructose Metabolism in NAFLD

One point of interest regarding fructose metabolism triggering NAFLD at the whole-body level is how intestinal and hepatic fructose metabolism is intertwined (Figure 2). The stable isotope tracing of ^13^C-fructose or glucose in mice has recently shown that the small intestine is the first-pass organ that catabolizes a majority of dietary fructose before fructose reaches the liver [3]. At low doses (0.25–0.5 g/kg), fructose is almost completely catabolized by the small intestine into glucose or other organic acids, such as lactate and alanine. This process reduces the amount of fructose reaching the portal blood and F1P production in the liver. However, at high fructose doses, the small intestinal capacity is overwhelmed and fructose spillover to the liver occurs. Importantly, the ability of the small intestine to clear the fructose is augmented by prior exposure to fructose or food consumption [3]. However, this small intestinal fructose catabolism may worsen NAFLD if it produces hepatotoxic metabolites from fructose.

To discern this possibility, intestine-specific KHK-C knockout mice were generated to abolish intestinal fructose catabolism [13]. Upon chronic fructose feeding, these mice showed increased hepatic lipogenesis, despite the fact that the mice drink less fructose water compared to the control. Consequently, they developed worsened fatty liver and hyperlipidemia, suggesting that intestinal fructose metabolism is protective. Consistent with this notion, intestine-specific KHK-C transgenic mice exhibited the opposite phenotypes, with reduced fructose spillover to the liver, less F1P generation, and lipogenesis [13]. However, chronic fructose feeding was not possible in the transgenic mice due to their fructose aversion phenotype, which seems to be related to increased F1P accumulation and consequent metabolic stress in the colon. 

To further test whether the speed of fructose intake influences intestinal fructose clearance, the effect of feeding bolus high-dose fructose (2 g/kg) versus split doses (0.5 g/kg × 4) on hepatic lipogenesis was compared [13]. Only high-dose fructose reached the liver and induced lipogenesis, suggesting that the rate of fructose consumption can determine fructose’s pathogenic impact on the liver. Therefore, the small intestine acts as a shield for the liver by preventing excessive fructose from reaching the liver. Consistently, a randomized controlled trial of 34 men found that the frequency of high-fat, high-sugar meals, rather than the size, leads to increased hepatic triglycerides and decreased insulin sensitivity [53].

Andres-Hernando et al. proved that hepatic fructose catabolism is the key cause of NAFLD [14]. Specifically, they investigated the different effects of KHK-C/A deletion in the liver, intestine, and whole body after chronic fructose drinking. The whole-body KHK-C/A knockout mice showed no NAFLD induction, largely due to the lack of fructose metabolism and decreased voluntary fructose drinking. However, in wild-type mice or mice with intestinal KHK-C/A knockout, NAFLD was developed. In contrast, mice with hepatic KHK-C/A knockout did not develop NAFLD because fructose cannot be catabolized by the liver and is thus mostly excreted into the urine. This study clearly indicates that fructose metabolism within the liver is the key driver of NAFLD. 

Why does intestinal and hepatic fructose catabolism lead to such different fates of fructose and metabolic consequences? One potential reason is the different capacity for lipogenesis [39]. The liver is designed to convert excess carbohydrate to fat in the post-prandial state. On the other hand, the primary role of the small intestine is to conduct nutritional absorption rather than anabolism. While intestinal lipogenesis has been suggested [54], it mainly involves the packing of dietary fatty acids to triglycerides in chylomicron particles rather than de novo lipogenesis (making nascent fatty acids from other nutrients). Indeed, there is no clear evidence of intestinal de novo lipogenesis in vivo [13]. Another difference between the liver and small intestinal fructose metabolism may be related to their different anatomy. Between the gut lumen and blood vessels that underly villi, there is only a single-cell layer of intestinal epithelial cells that express KHK. This structure limits the fructose catabolic capacity, which can be saturated by excessive fructose [55]. On the other hand, numerous KHK-expressing hepatocytes exist between the portal vein and hepatic vein in the liver, which largely prevents fructose spillover to the peripheral system [56] (Figure 3). 

## 5. Fructose Metabolism by Other Host Organs

Although most dietary fructose is catabolized by the small intestine and the liver, some fructose escapes to the systemic blood and can be catabolized by other organs, such as the heart, pancreas, kidneys, skeletal muscle, and adipose tissues [16,57,58,59,60,61,62,63]. GLUT5 expression has been reported in the brain, kidney, adipose tissue, and skeletal muscle tissue [57]. Additionally, GLUT2 is highly expressed in the kidney [57]. While the expression of these fructose transporters and KHK in extra-hepatic organs suggests their capacity to catabolize fructose, quantitatively, how much of the fructose, either from the diet or endogenous production, is catabolized by these organs and their metabolic consequence is relatively unknown. 

It has been reported that fructose transporters or catabolizing enzymes are induced by diseases and mediate some pathological phenotypes. Mirtschink et al. found that the activation of hypoxia-inducible factor 1α (HIF1α) in pathogenic hearts activates splicing factor 3B subunit 1 (SF3B1), which shifts alternative splicing from KHK-A to KHK-C [58]. KHK-C then increases cardiac fructose metabolism and induces hypertrophic growth via ATP depletion, which were partially rescued in KHK knockout mice [58]. Interestingly, this study was performed in the absence of dietary fructose, suggesting that endogenous fructose is catabolized by the heart. It is unclear whether endogenous fructose can drive sufficiently high fluxes to deplete ATP in the heart, which is the most active ATP-generating/consuming organ. Further studies with dietary fructose feeding or stable isotope tracing to measure fructose catabolizing fluxes are required to answer these questions. 

Interestingly, KHK-C is also highly expressed in the kidneys [16]. Consistently, fructose catabolism in the kidney is associated with chronic kidney disease [59,63]. Gersch et al. fed rats diets of either 60% fructose, 60% glucose, or normal chow. Compared to other rats, the rats fed 60% fructose exhibited pathological kidneys with more tubular dilation, glomerulosclerosis, tubular atrophy, myofibroblasts, and interstitial inflammation, eventually leading to hypertension, which is a symptom observed in patients with advanced chronic kidney disease [59]. Another study by Roncal-Jimenez et al. echoed these results [63]. Both wild-type and KHK knockout mice were fed a typical diet with low doses of fructose chronically. The wild-type mice experienced glomerular changes and kidney damage, whereas the KHK knockout mice were protected. 

There are also reports showing the detrimental effect of excessive fructose exposure to various organ-specific cell types. Bartley et al. examined the effects of fructose on pancreatic β-cells [60]. Although fructose itself did not cause insulin exocytosis due to the β-cells’ lack of GLUT5, the constant exposure to fructose resulted in the hyper-reactivity of pancreatic β-cells to glucose, with ATP depletion [60]. In adipose tissue, insulin increases fructose uptake and catabolism by the fat cells [61]. Consistently, in diabetic patients with adipose insulin resistance, this fructose uptake was decreased [61]. On the other hand, in the isolated human skeletal muscle, Zierath et al. found no activation of fructose usage by insulin, although fructose conversion to lactate was tripled, with increasing fructose levels from 0.1 to 0.5 mM [62]. These studies suggest that high-dose fructose can be metabolized by a variety of organs and tissues, while further investigations are needed to identify the physiological relevance in vivo where circulating fructose levels are significantly lower than the levels of fructose used in these conditions. 

## 6. Microbial Fructose Metabolism in NAFLD

One emerging development in the field of fructose metabolism is the role of the gut microbiota. Recent breakthroughs in meta-genome sequencing and metabolomics technologies have identified the link between differential gut microbiota subpopulations and their metabolic products in the context of NAFLD [28,64,65,66]. Because the small intestinal fructose absorption is limited [38], high-dose fructose readily reaches the large intestine and induces bacterial fermentation [67,68,69,70]. A study of 15 healthy adults found that more than half of the participants reported some gastrointestinal symptoms at a dosage of 25 g fructose and more than two thirds reported distress at a dosage of 50 g fructose [68]. Interestingly, fructose ingested as sucrose or in conjunction with glucose significantly increased the absorption capacity [69], consistent with the notion that glucose facilitates intestinal fructose absorption [70]. Furthermore, heightened H_2_ production rates associated with malabsorption were observed in some healthy individuals, suggesting that differential fructose metabolism capabilities may lead to gastrointestinal symptoms in some fructose-sensitive individuals [69,71]. 

Once fructose reaches the large intestine, the gut microbiota catabolize fructose into various metabolites, including short-chain fatty acids (SCFAs: acetate, butyrate, and propionate), TCA cycle intermediates, and amino acids [3]. However, an important question is which gut microbiota species catabolize fructose and whether this process is related to NAFLD. It is possible that all gut microbiota species are capable of catabolizing fructose to some extent and contribute to the disease. Microbiota do not have KHK and instead use hexokinases to phosphorylate fructose into fructose-6-phosphate for subsequent glycolysis [72]. Alternatively, fructose may feed the intestinal microbiome subpopulations, in which bacteria with a high catalytic ability either outcompete or form cooperative relationships with those having no fructose metabolic capabilities [73]. Either way, a chronic fructose intake likely changes the gut microbial ecosystem [74], which can trigger NAFLD in various ways. 

The most well-studied microbiota species in the context of NAFLD are *Bacteroidetes* and *Firmicutes*, yet there is still controversy over which species is more detrimental. For example, the fecal sequencing of rats on high-fructose diets displayed a marked increase in *Bacteroidetes:Firmicutes* ratios [70]. On the other hand, a similar study with pregnant rats suggested decreased *Bacteroidetes* levels [65]. In humans, an overrepresentation of several species of *Firmicutes* was found in a study involving obese individuals [69]. These results were corroborated by an epidemiological study of a cohort of overweight/obese Hispanic teenagers, where a strong negative correlation between fructose consumption and *Firmicutes*, particularly *Eubacteria elegens*, was found [75]. However, it is worth noting that generalization from a cohort to the wider population remains difficult due to the effects of the genetic makeup, environment, and culture on the microbiome. It is also evident that not all population shifts in the microbiome share a causal relationship with fructose metabolism, as late-stage microbiome changes may be more associated with systemic effects (e.g., obesity and insulin resistance), rather than being a direct response to the higher-fructose diet. 

While it is important to identify which microbiota species catabolize fructose and cause NAFLD, an equally crucial question is which microbial metabolic products cause hepatic lipogenesis and inflammation in NAFLD. Indeed, whilst the connection between differential *Bacteriodetes:Firmicutes* populations and hepatic lipogenesis is still unclear, the amount of SCFAs produced by both bacterial phyla were significantly greater in obese patients than in lean patients [76]. Microbiota-derived SCFAs play versatile roles in processes including colonic epithelial cell differentiation, epigenetic modifications (e.g., histone acetylation), hepatic lipogenesis, and G protein-coupled receptors (GPCRs)-mediated signal transduction [77,78,79,80,81]. In particular, SCFAs have been linked to lipogenesis via histone deacetylase inhibition and GPCR41/43 activation, the latter of which has implications in NAFLD [80,82,83]. 

Quantitatively, acetate is the most abundant SCFA, with an-order-of-magnitude higher levels than the other SCFAs. It is also one of the metabolites with a high turnover rate in circulation [84]. Recently, acetate has been shown to be a crucial carbon source made from fructose by gut microbiota for hepatic acetyl-CoA and fatty acids [41]. Zhao et al. unexpectedly found that liver-specific ATP citrate lyase (ACLY)-knockout mice still developed NAFLD phenotypes upon high-fructose diet feeding, even though these mice were not able to use cytosolic citrate for lipogenesis. Using antibiotics treatment and the liver-specific knockdown of Acyl-coenzyme A synthetase short-chain family member 2 (ACSS2), which is the essential enzyme for acetate catabolism, they showed that depletion of the microbiota or suppression of ACSS2 dramatically reduced fructose carbon incorporation into hepatic acetyl-CoA and fatty acids [41]. Importantly, this effect was independent of the fructose-induced upregulation of hepatic lipogenic genes, suggesting a dual action of fructose as a signaling molecule and as a carbon source for lipogenesis via acetate. This finding was also consistent with the high conversion of intra-cecal infused stable isotope-labeled acetate and butyrate into hepatic fatty acids [85]. Further supporting this notion, hepatic lipid accumulation in rodents fed high-fructose diets was rescued by both antibiotic treatment and a fecal transplant from healthy rodents [10,86,87]. As such, antibiotics may be a viable treatment option for NAFLD. However, long-term antibiotic treatment is not ideal and the chronic depletion of beneficial microbiota can cause other symptoms. Therefore, targeted therapies will be required, which will be discussed in the next section. In addition to SCFAs, untargeted metabolomics are needed to discover novel microbial metabolites associated with NAFLD.

Another dominant theory for the occurrence of fructose-elicited NAFLD through gut microbiota is an increased gut leakiness and the consequent spillover of microbial products that trigger liver inflammation. Fructose that reaches the colon can induce osmotic shock and loosen the gut epithelial tight junctions. Indeed, several studies have reported a reduced expression of tight junction proteins in rodents fed high-fructose diets [88]. This leads to an increased delivery of bacterial endotoxins such as lipopolysaccharide (LPS) to the portal circulation and thereby induces hepatic inflammation [89]. Additionally, dysbiosis of microbiota induced by a chronic fructose intake may remodel the gut immune system toward an increased recruitment of macrophages and T cells that release inflammatory cytokines such as TNFα and IL-6 to hepatocytes [10,86,89,90]. Mice lacking inflammatory receptor TLR4 show a markedly lower onset of steatosis when fed a high-fat diet [91]. Inflammatory signals may either directly activate hepatic lipogenesis or induce hepatic insulin resistance to indirectly induce lipid accumulation [92]. A recent study by Todoric et al. found increased gut leakiness and endotoxemia in mice fed a chronic high-fructose diet [9]. These microbiota-derived Toll-like receptor (TLR) agonists trigger a hepatic inflammatory response and subsequent downstream cytokine pathways, leading to lipogenesis induction and fatty liver development. These results further tighten the connection between hepatic inflammation due to microbial TLR4 agonists and fatty liver [93].

## 7. Future Perspectives: Developments in the Prevention of Fructose-Induced NAFLD

Small molecules can be effective therapeutics in the treatment of fructose-induced NAFLD and related comorbidities. One potential category of drugs includes inhibitors of KHK or lipogenic enzymes, such as Acetyl-CoA carboxylase (ACC). KHK in particular has been singled out as a druggable target because of its known mutation in humans with a benign phenotype—essential fructosuria [94]. This enzymatic dysfunction results in fructose remaining in its non-phosphorylated form in plasma and at higher levels of excretion in urine [95]. Mice lacking both KHK-A and KHK-C showed a lower body weight, lipid accumulation, and lipogenesis than in control mice under high-fat, high-sugar diets [96]. Further analysis by Lanaspa et al. suggests that KHK loss-of-function also protects against endogenous fructose production, in which the conversion of serum glucose at high concentrations to fructose contributes to fatty liver, even without an excessive calorie intake [97]. Therefore, there is both a molecular and genetic basis on which KHK inhibitors may prove to be an effective treatment against hepatic NAFLD.

The in vitro analysis of human KHK-C inhibitors points to pyrimidinopyrimidine compounds as selective and highly affinitive inhibitors, with some species demonstrating both oral bioavailability in rats and efficacy in cellular functional assays [95]. While promising, it was also noted that there was a high rate of clearance (C_max_ = 0.16 µM), bringing into question whether pyrimidinopyrimidine inhibitors can reach therapeutic concentrations in vivo. Another in vitro study of KHK inhibition involving an indazole series also demonstrated ATP-binding capabilities with good pharmacokinetics in rodent models [98]. Huard et al. administered a potential pyridine inhibitor for rat and human KHK to Sprague Dawley rats, resulting in a nonlinear decrease in F1P in the liver and kidneys in mere minutes after bolus fructose feeding, with a reported hepatic ED_50_ of 30.0 mg/kg [99]. Based on these in vitro and animal studies, a KHK inhibitor has been applied in a Phase II clinical trial, exhibiting promising effects so far.

Small molecule inhibitors targeting ACC, which is a rate-limiting enzyme that modulates the conversion of acetyl-CoA to malonyl-CoA, have also been posed as therapeutic agents against hepatic lipogenesis. However, there is some contention on the significance of hepatic lipogenesis in human NAFLD: Most accumulated triglycerides in hepatocytes are synthesized from adipose tissue rather than through hepatic lipogenesis [100] and the results of rodent in vivo studies must be contextualized by a higher absolute rate of lipogenesis in rodents than in humans [101]. Nevertheless, it is also evident that there is some connection between NAFLD and hepatic lipogenesis. In clinical studies, for example, lean patients were shown to have lower hepatic lipogenesis than their obese counterparts [100,102]. In human patients with non-alcoholic steatohepatitis, hepatic lipogenesis accounts for up to 38% of liver palmitate [103]. The liver-specific inhibition of ACC1 and ACC2 with small molecules has been shown to decrease hepatic triglyceride levels and lipogenesis in NAFLD patients [104]. However, plasma triglycerides rose in these individuals, likely due to the activation of the lipogenic transcription factor SREBP-1c downstream of the ACC inhibition pathway. Therefore, it is evident that there is complex interplay between the suppression of lipogenic liver enzymes and systemic triglyceride formation.

In addition to drug development for fructose-elicited NAFLD, changing one’s lifestyle can be an effective preventive strategy. There is evidence of a differential response to the form of fructose ingested and NAFLD development [105,106,107,108,109]. Johnson et al. analyzed the effect of HFCS and sucrose on NAFLD occurrence and insulin resistance in Iberian pigs [110]. They found that both the sucrose and HFCS diet increased the body weight and induced steatosis. Yu et al. analyzed the effect of HFCS and sucrose on energy-regulating hormones as a randomized human trial [111]. In total, 138 patients were administered low-fat milk with either HFCS or sucrose added and were then analyzed over ten weeks. The sugar levels were based on the 25th, 50th, and 90th percentiles of sugar consumption. The impact was indistinguishable between HFCS and sucrose, with both groups showing increased leptin, insulin, and triglyceride levels, but no change in glucose or uric acid. This is consistent with the notion that sucrose is almost immediately cleaved to glucose and fructose by sucrase. Therefore, although sucrose is a natural product, its metabolic impacts are similar to HFCS.

On the other hand, several groups have reported a differential effect of liquid-form versus sold-form fructose on health. Mice that were fed a liquid sucrose diet had higher hepatic triglycerides than their isocaloric solid sucrose-fed counterparts [108]. In a 6-year longitudinal study of European children and adolescents, a stronger link between liquid sucrose and BMI and waist circumference was established than that between these physiological modifications and either sugar, sucrose, or solid sucrose [109]. In particular, the connection between liquid sucrose and higher waist circumference/abdominal fat was attributed to insulin resistance rather than the total caloric intake, perhaps due to a more robust insulin response from liquid sucrose than solid sucrose [105,106]. This rapid increase in insulin levels has been linked to both insulin resistance and subsequent visceral fat deposition. In this regard, reducing the intake of liquid-form fructose is a recommended dietary intervention for preventing NAFLD.

## 8. Conclusions

Advanced applications of biological and chemical technologies in animal models have provided new insights into the contribution of whole-body fructose metabolism and gut microbiota to fructose-induced NAFLD. Most of these recent findings were unrecognized by tissue culture studies, demonstrating the importance of investigating metabolism and disease in the context of complex inter-organ communication. While these animal model studies have greatly expanded our understanding of in vivo fructose metabolism and its causal effects on NAFLD, relevance in humans represents the remaining unanswered question. Studying large animal models similar to humans in terms of anatomy, metabolism, and pathophysiology may be an option. With several promising clinical trials targeting NAFLD driven from animal studies, continuous investigations and collaborations by the scientific community will be key to understanding the pathology of NAFLD and its underlying molecular mechanisms for clinical diagnosis and treatment.

## Figures and Tables

**Figure 1 biology-09-00405-f001:**
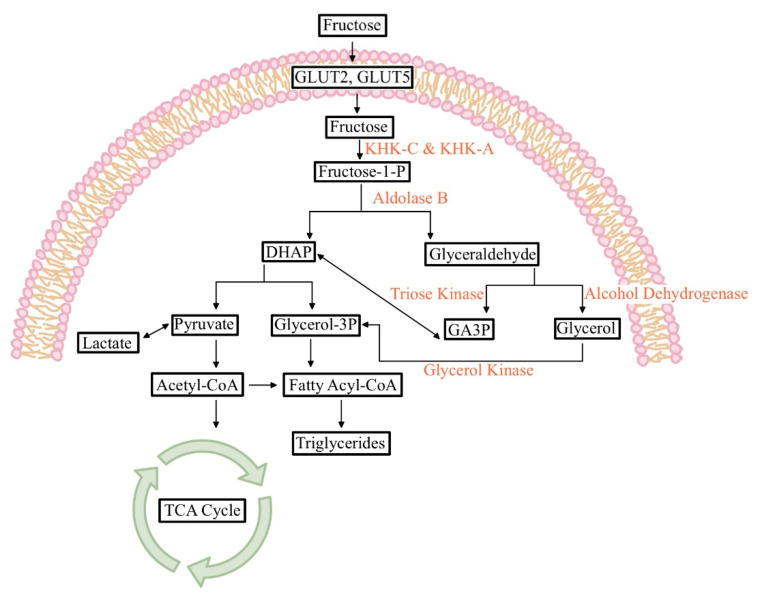
Fructose metabolism pathway. Fructose taken up by Glut2 or Glut5 is subsequently phosphorylated by ketohexokinase (KHK) into fructose-1-phosphate. Aldolase B then cleaves it to three carbon units, dihydroxyacetone phosphate (DHAP), and glyceraldehyde. Glyceraldehyde becomes glyceraldehyde-3-phosphate (GA3P) to enter glycolysis or becomes glycerol-3-phosphate to provide the glycerol backbone of newly synthesized lipids (e.g., triglycerides).

**Figure 2 biology-09-00405-f002:**
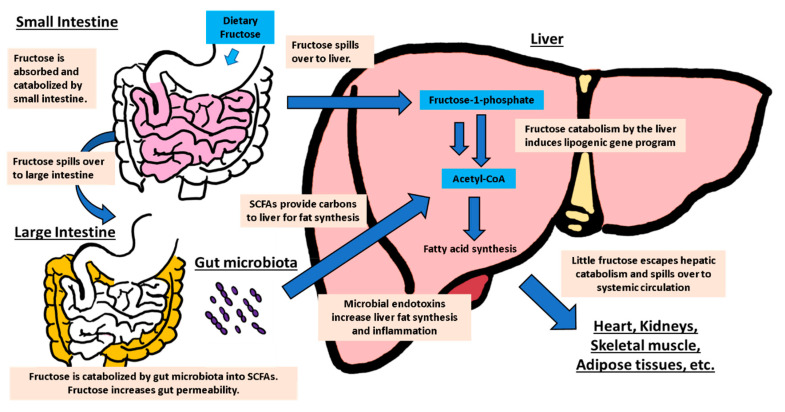
Contribution of whole-body fructose metabolism to non-alcoholic fatty liver disease (NAFLD). Physiological doses of dietary fructose are mostly catabolized by the small intestine. However, high-doses or rapid consumption (e.g., liquid-form) results in fructose spillover to both the liver and the large intestine. In the liver, uncontrolled fructose catabolism induces hepatic lipogenesis. In the large intestine, fructose feeds gut microbiota, generating short-chain fatty acids (SCFAs) as a carbon source of hepatic lipogenesis. Fructose also increases the gut permeability, enhancing hepatic inflammation via endotoxins. Most dietary fructose is cleared by the intestine and the liver and very little reaches other organs.

**Figure 3 biology-09-00405-f003:**
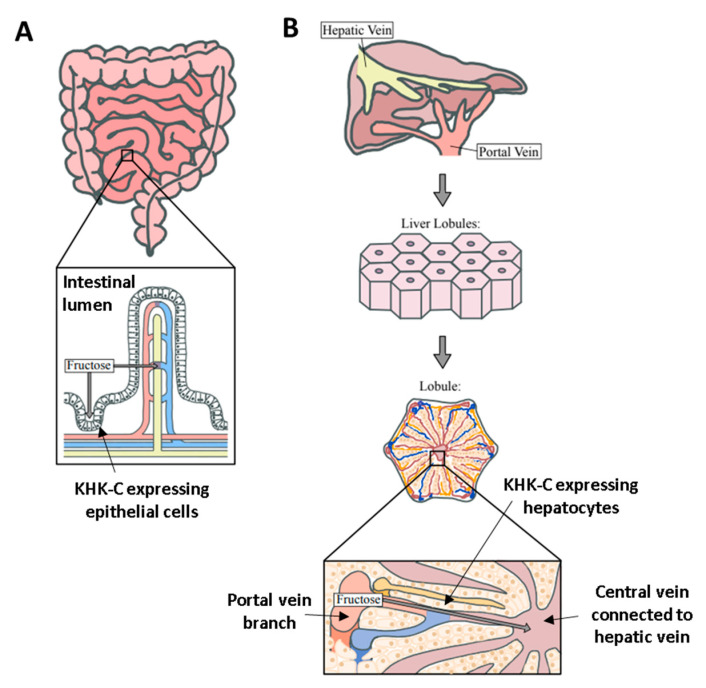
Anatomical difference between the liver and intestine for fructose catabolism. (**A**) In the intestine, there is a single-layer of KHK-C expressing epithelial cells between the intestinal lumen and blood vessel (fructose diffusion is perpendicular to the cell layer). This structure limits the fructose catabolic capacity when the delivered fructose dose is high. (**B**) In contrast, in the liver, there are numerous KHK-C-expressing hepatocytes lining the portal-to-hepatic circulation. Hepatocytes are also metabolically highly active to efficiently assimilate fructose carbons via gluconeogenesis and fat synthesis.

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
