# Peer review of "Organismal Fructose Metabolism in Health and Non-Alcoholic Fatty Liver Disease"

_biology, 2020, doi:10.3390/biology9110405_

Round 1

Reviewer 1 Report

Non-alcoholic fatty liver disease (NAFLD) has been alarmingly increased, and excessive consumption of fructose is a dietary risk factor of NAFLD. This review summarized organismal fructose metabolism distributed among various organs and the potential contribution of organ-specific fructose metabolism to NAFLD. A few comments were provided as bellow:

  1. Since the manuscript mainly focused on fructose metabolism in NAFLD, it is better to modify the title to fit the context. Fatty liver disease also included alcoholic liver disease, which the author did not discuss.
  2. Is there a title missed for section 2?
  3. A schematic figure indicating key fructose transporter and the molecular signaling in NAFLD will help to understand the context of section 2.1 and 2.2
  4. Should label two figures in Figure 2 as Figure 2A and Figure 2B instead of “left” and “right”?
  5. How is the clinical relevance between fructose consumption and NAFLD?

Reviewer 2 Report

In this review, the authors summarise the role that Fructose absorption and metabolism plays in lipogenesis and the pathogenesis of NAFLD/NASH. The review, in particular, covers findings from genetically modified animal models and gut microbiota. This topic has gained significant momentum in recent times, with many studies demonstrating that fructose can be a key contributor to NAFLD. The review is well written, bar some minor grammar, covering some key aspects and limitations of previous studies. The review is of great interest and worthy of publication, pending some minor modifications.

  1. Can the authors please provide some more information in the introduction for sources of fructose? Although the authors go into detail about liquid fructose v solid, more information about naturally occurring fructose in fruits for example (Line 30), and whether these also constitute NAFLD risk. Further to this, the authors first mention endogenous fructose at Line 245, which could also be introduced and clarified earlier in the review. Presently, it is introduced as a throw away line, but then it is commented on in sections 2.4 and 2.6.
  2. Further to question 1, is the source of fructose relevant for the progression of NAFLD, ie derived from corn syrup vs traditional sources?
  3. Can the authors provide some more information on observed phenotypes? For example, Line 57: Is it known why the intestine becomes distended? Is fructose important here? This is also observed Line 207. Line 59: humans with Glut 5 mutations.
  4. Can the authors briefly describe the limitations of the human study, Line 124, to further support the review.

Reviewer 3 Report

The authors did and nice job of reviewing and raised very good points. Very nice as well is the overview of knock-out genetic models employed for the study high fructose diet. Following there are few suggestions to improve the review.

Paragraph 1

I would suggest extending a bit more the concept that, at least for a decade, the cholesterol, among other types of fat, has been indicated as the only culprit for non-alcoholic fatty liver disease.  

Paragraph 2.1 (I would add a figure that summarize the fructose metabolism into intestine and liver)

From line 50 to 53, expand a little bit more the difference in the transport of glucose and fructose during the intestinal absorption and the role of GLUT5 and Glut2.

From line 54 to 66, I would suggest grouping all the information (phenotype and location) together for one transporter and then the other. Moreover, add some details (or reference) about the phenotype in human of the mutation for Glut5 that the authors mentioned (line 59) and then left there.

Line 67, the authors state that the other fate for the fructose that does not go into the portal circulation is catabolized by ketohexokinase (KHK), which 68 phosphorylates fructose into fructose-1-phosphate (F1P). I would add also where and by what cells.

Line 69, please specify which transporter is greatly expressed in the liver, small intestine, pancreas, and which on is more broadly expressed in multiple tissues.

From line 83, I would delete this sentence: “Fructose catabolism in the intestine is also crucial for induction of fructose-related genes upon 83 chronic fructose exposure as an adaptive mechanism”. It is too vague and break the connection with following part where the authors still discuss about KHK. Moreover, I would mention which are the fructose-responsive genes studied by Mavrias et al.

Line 88 I would rephrase like that: Kim et al. proved that ChREBP knockout mice have suppressed Glut5, KHK, and other fructolytic and gluconeogenic enzyme expression even after chronic fructose feeding

Line 117 Since the authors did a great job in grouping the literature about fructose feeding experiments in rodents, it would be nice to specify the range of the “high dose” used in those studies. This would help the reader to make an idea about the quality of the studies present in the literature, since many papers that study the effect of one specific diet or another feed the animals “ad libitum”.

Line 124 add “triglyceride” to the sentence: the examination of the serum and liver.

Line 142 Within the liver, the fructose in the liver can enter in multiple pathways dependently whether its ingestion is chronic or acute. Many human studies focused on high intake of either fats, sugars, or proteins for short period of time which activated completely different pathways than their chronic ingestion. This approach does not give the time the adapt to change in the diet letting the authors drawing incorrect conclusion. I would suggest adding some other well-known pathways where the fructose can be metabolised (e.g. de novo lipogenesis, PPP pathway and so on see Jegatheesan, Nutrients 2017 and De Chiara, Nutrients 2019).

Line 207. Interesting point raised by the authors is that only when a single high dose rather than multiple small doses of fructose overwhelms the small intestine reaching the liver. An interesting analogy is between their work and the paper of Koopman, Hepatology 2014. It would be nice if the authors could ad a comment on that.
